# Epidemiological Survey on Tick-Borne Pathogens with Zoonotic Potential in Dog Populations of Southern Ethiopia

**DOI:** 10.3390/tropicalmed8020102

**Published:** 2023-02-03

**Authors:** Hana Tadesse, Marika Grillini, Giulia Simonato, Alessandra Mondin, Giorgia Dotto, Antonio Frangipane di Regalbono, Bersissa Kumsa, Rudi Cassini, Maria Luisa Menandro

**Affiliations:** 1Arba Minch Agricultural Research Center, Southern Agricultural Research Institute, Arba Minch P.O. Box 2228, Ethiopia; 2Department of Veterinary Parasitology and Pathology, Addis Ababa University, Bishoftu P.O. Box 34, Ethiopia; 3Department of Animal Medicine, Production and Health, University of Padova, 35020 Legnaro, PD, Italy

**Keywords:** dogs, tick-borne pathogens, *Anaplasma*, *Ehrlichia*, Rickettsia, Borrelia, piroplasm, Ethiopia

## Abstract

Dogs are known to host several tick-borne pathogens with zoonotic potential; however, scant information is available on the epidemiology of these pathogens in low-income tropical countries and in particular in sub-Saharan Africa. With the aim of investigating a wide range of tick-borne pathogens (i.e., *Rickettsia* spp., *Anaplasma* spp., *Erhlichia* spp., *Borrelia* spp., *Hepatozoon* spp. and *Babesia* spp.), 273 blood samples were collected from dogs in selected districts of Ethiopia and analyzed by real-time and/or end-point PCR. The results of the study showed that *Hepatozoon canis* was the most prevalent pathogen (53.8%), followed by *Anaplasma phagocythophilum* (7.0%), *Babesia canis rossi* (3.3%), *Ehrlichia canis* (2.6%) and *Anaplasma platys* (2.2%). Furthermore, five samples tested positive for *Borrelia* spp., identified as *Borrelia afzelii* (*n* = 3) and *Borrelia burgdorferi* (*n* = 2), and two samples for *Rickettsia* spp., identified as *Rickettsia conorii* (*n* = 1) and *Rickettsia monacensis* (*n* = 1). The finding of *Anaplasma phagocythophilum* and different species of the genera *Borrelia* and *Rickettsia* with zoonotic potential was unexpected and alarming, and calls for further investigation on the roles of dogs and on the tick, species acting as vector in this specific context. Other pathogens (*Hepatozoon canis*, *Babaesia canis rossi*, *Anaplasma platys*, *Ehrlichia canis*) are already known to have an important impact on the dogs’ health but have minor zoonotic potential as they were rarely or never reported in humans. Dogs from rural areas were found to be at higher risk for different pathogens, probably due to the presence of other wild canids in the same environment. The findings of the present study contribute to a better knowledge of the epidemiology of tick-borne pathogens, which is relevant to human and animal health.

## 1. Introduction

Tick-borne diseases (TBDs) are a major issue in the African context, but the epidemiology of the tick-borne pathogens linked mainly to dogs have been poorly investigated in the African continent. On the contrary, these pathogens have received huge attention in other areas of the world, being included in the broader umbrella of canine vector-borne diseases (CVBDs). These infections have worldwide distribution, but climatic and environmental changes and an increase in the mobility of people and animals have contributed to the spread of vectors and pathogens, leading to increasing morbidity and mortality in animals and humans [1]. The increase in dog populations and their co-habitation with humans in urban and rural environments pose new threats to human health [2,3]. If not properly treated, dogs are more likely to be exposed to pathogens transmitted by vectors and to become competent reservoirs of several pathogens [4,5]. Canine vector-borne diseases have a major impact on animal health and welfare and, in many cases, on human health as well. The effective surveillance of CVBDs requires clear and exhaustive knowledge on their distribution in different areas of the world [6,7,8]. Recent investigations indicate worldwide escalating problems associated with arthropods and vector-borne diseases (VBDs) caused by the pathogens they transmit [5,9,10].

In tropical countries, arthropods are very common and widely distributed due to the presence of a highly conducive climate and a lack of prevention and control measures [11]. Blood-feeding arthropods (e.g., ticks, fleas, mosquitoes and sand flies) act as vectors and reservoirs for several agents of emerging and re-emerging infectious diseases of medical and veterinary importance [12]. Among these, ticks are widespread in most species of domestic animals and throughout different environments in Africa, predisposing this continent to a huge potential diffusion of TBDs. However, little information is available on canine TBDs for most African countries. The bacterial pathogens *Rickettsia* spp., *Anaplasma* spp. and *Erhlichia* spp. are usually considered the most important TBDs affecting dogs [13,14,15,16,17], but other bacterial pathogens and the protozoan parasites *Babesia* spp. and *Hepatozoon* spp. are also reported in some countries [18,19,20]. Most of the above-mentioned pathogens have zoonotic potential and represent a threat to human health [21].

Like many African countries, the rate of urbanization in Ethiopia is increasing rapidly, as is the number of dogs kept as companion animals, resulting in high densities of human and dog populations in urban areas. Consequent to uncontrolled populations of dogs living near increasing densities of human populations, the effective control of canine-originated diseases is an extremely challenging task. Despite the presumably large number of dog populations in Ethiopia, research on canine vector-borne pathogens is scanty or absent and poor attention is given to diagnosis, prevention and control programs. Previously, a study was conducted by Kumsa et al. [22] in central Oromia, Ethiopia on fleas collected from dogs to detect *Rickettsia felis*, but no molecular studies on canine blood samples were performed. Knowledge of the determinants of TBDs in dogs is essential in assessing the risk of transmission to humans and planning effective prevention and control in the country. Therefore, the present study aimed to investigate the occurrence and prevalence of bacterial and protozoal tick-borne pathogens using molecular methods in owned dogs and to determine the associated risk factors in selected districts of Southern Ethiopia.

## 2. Materials and Methods

### 2.1. Sampling

Gamo Zone is located in Southern Nations, Nationalities, and People Region, between 5°55′ N and 6°20′ N latitude and between 37°10′ E and 37°40′ E longitude, and it covers an area of 6735 km^2^. Elevation ranges between 600 and 4207 m above sea level (masl), and three agroecologies can be identified: highland (>2300 masl), midland (1500–2300 masl) and lowland (<1500 masl). The mean annual rainfall ranges from 200 mm to 2000 mm, and the annual average temperature ranges from 15 °C to 28 °C, with decreasing values at increasing altitude. Most of the natural vegetation consists of woodland and savannas, but afro montane forests are found in the highlands.

Sampling was carried out between November 2020 and January 2021 in four districts of the Gamo zone in Southern Ethiopia: Arba Minch town, Chencha town, Arba Minch zuria and Gerese. These districts were selected by the Gamo zone administration, being the ones with higher dog populations, and households to be visited for sample collection were opportunistically identified. An expected sample size of 276 animals was defined at the beginning of the study, in order to be able to estimate the prevalence values of concerned pathogens with a 6% accepted error (95% confidence level), given an expected prevalence of 50% and an infinite population. Information on sex, age (i.e., young, adult), district of provenance, location (i.e., urban, rural), lifestyle (i.e., only outdoor, mixed indoor–outdoor) and agroecology (i.e., highland, midland, lowland) was recorded for each dog.

Whole blood samples were collected from the cephalic vein and were applied (100 μL) to classic Flinders Technology Associates (FTA™) Nucleic Acid Collection Cards (Whatman^®^, Maidstone, UK), then air dried, coded and stored in sealed FTA pouches with a silica gel desiccant (Sigma Aldrich, Co., Life Sciences, St. Louis, MO, USA) until analysis. Blood samples were collected from privately owned dogs after the owner’s verbal consent to participate in the study had been given. Most of the owned dogs are usually left to free roaming during the day in the proximity of their home. Stray dogs were not included in the sample, since their safe containment and management is extremely difficult in this context. Only asymptomatic dogs were included in the study, and each dog was carefully observed for the presence of ectoparasites (including ticks).

FTA cards with blood samples were transported to the Laboratory of Parasitology of the University of Padova, Legnaro (PD), Italy, as per authorization of the Italian Ministry of Health (Prot. N. 0002711, 22 April 2021). The study protocol was approved by the Animal Ethics Committee of Addis Ababa University, College of Veterinary Medicine and Agriculture (Agreement No. 17/168/550/2009).

### 2.2. Laboratory Analysis

DNA was extracted from dried blood spots found in the FTA cards that were punched (diameter of punches: 6 mm) and one punch was put into an Eppendorf tube labeled with the same number reported on the FTA cards. The samples were processed using NucleoSpin™ Tissue extraction kit (Macherey-Nagel, Düren, Germany) according to the manufacturer’s instructions. The purified DNA was stored in the freezer at −20 °C until molecular analysis execution.

The detection of bacterial pathogens was performed using specific end-point PCR assays. All reactions were performed using a Biometra T Advanced thermocycler (Analytic Jena, Gottingen, Germany), Phire Hot Start II PCR Master Mix (Thermo Scientific, Vilnius, Lithuania) and 0.3 μM of each primer. Positive and negative controls have been included in each run. The presence of *Anaplasma phagocythophilum* (*A. phagocythophilum*), *Anaplasma platys* (*A. platys*), *Erhlichia canis* (*E. canis*) and *Borrelia burgorderfi* sensu latu (*B. burgorderfi* s.l.) was checked with primers targeting a portion of the gene encoding the groEL heat shock protein [23,24,25], while *Rickettsia* sp. DNA was detected using the primer pair described by Regnery et al. [26] which targets a portion of the citrate synthase gene (*gltA*). Primer sequences and expected product length have been summarized in Table 1. Amplification products were visualized by electrophoresis on 2% agarose gels stained with SybrSafe DNA Stain (Invitrogen, Thermo Scientific, Carlsbad, CA, USA).

For the detection of protozoal pathogens (*Babesia* spp. and *Hepatozoon* spp.), DNA extracts were initially analyzed by real-time PCR using the QuantiNova SYBR^®^ Green PCR Kit (QIAGEN Group, Hilden, Germany) with primers previously described by Tabar et al. [27] targeting the 18S-rRNA gene spanning the V4 region (Table 1). The assay was performed in the Roche LightCycler^®^96 thermocycler (La Roche Ltd., Basel, Switzerland) with the following amplification cycle: incubation at 95 °C for 2 min, followed by 45 cycles of amplification steps at 95 °C for 5 s and 60 °C for 10 s, concluding at 95 °C for 10 s, 65 °C for 1 min and 97 °C for 1 s. The melting curve analysis was performed by continuously monitoring the fluorescence while decreasing the temperature from 95 °C to 65 °C. Fluorescence specificity and the identification of target pathogens in dogs were achieved through the melting temperature curve analysis. The melting temperature of the more common canine piroplasms of the two genera *Hepatozoon* and *Babesia* (respectively, *H. canis* and *B. canis*) were identified and compared with those of other relevant piroplasms, potentially present in other domestic or wild hosts [28]. The melting temperature for *H. canis* (79/79.5 °C) was determined using an isolate of *H. canis* as positive control obtained through preliminary analysis of 20 samples of the present study. The melting temperature for *B. canis* (81.5 °C) was determined using as positive control a *B. canis canis* field sample provided by the Istituto Zooprofilattico Sperimentale delle Venezie (Legnaro, Italy). Positive (i.e., DNA of sequenced field samples) and negative (no DNA added) controls were added in each PCR reaction and all samples were tested in duplicate. Selected samples that resulted positive at real-time PCR analysis for *H. canis* and *B. canis* were submitted to end-point PCR targeting a 373 bp fragment of the 18S-rRNA gene using the same primers as before (Table 1). End-point PCR was performed using Invitrogen Taq DNA polymerase (Thermo Fisher Scientific Inc., Waltham, MA, USA), following the manufacturer’s instructions.

Amplified PCR products from positive samples were purified using EXOSAP-it^®^ (ExoSAP-IT™ PCR Product Cleanup, Thermo Fisher Scientific Inc., Waltham, MA, USA) according to the manufacturer’s instructions. Purified products were sent to Sanger bidirectional sequencing at Macrogen Spain (Madrid, Spain) or at StarSEQ^®^ GmbH facilities (Mainz, Germany), using the same primers used in the PCR assays. The nucleotide sequences were assembled and edited using ChromasPro v.2.1.8 (Technelysium Pty Ltd., Brisbane, Australia) and compared to those deposited in GenBank^®^ using BLAST (https://blast.ncbi.nlm.nih.gov/Blast) (accessed on 25 January 2023).

### 2.3. Data Analysis

Prevalence values and their 95% confidence intervals (95% CI) were calculated using the Clopper–Pearson exact method for each pathogen identified. Sex (i.e., male, female), age (i.e., young: 0–24 months, adult: ≥25 months), location (i.e., urban, rural), lifestyle (i.e., only outdoor, mixed indoor–outdoor), agroecology (i.e., highland, midland, lowland) and presence of ticks were used as variables to calculate proportions for all pathogen species sufficiently prevalent and summarized descriptively. The Pearson Chi-square test, or the Fisher’s exact test when more appropriate, was used to compare proportions for all identified pathogens, and, when more than one factor significantly associated with a specific parasite resulted, a multivariable logistic regression model was used to evaluate the association between the potential risk factors and that species. The model fitness was assessed by the Hosmer–Lemeshow goodness-of-fit test. A probability *p*-value < 0.05 was regarded as statistically significant. Analyses were conducted using the software IBM SPSS Statistics 27.

## 3. Results

### 3.1. Dog Population Description

A total of 273 blood samples were collected from dogs and analyzed molecularly, moving very close to the expected sample size number. Unfortunately, three samples were not well maintained after international transport and therefore not exploitable. Most of the sampled animals (n = 141; 51.6%) came from Arba Minch town, followed by Arba Minch zuria (n = 85; 31.1%), Gerese district (n = 36; 13.2%) and Chencha town (n = 11; 4%). Dogs lived in urban areas (64.5%), more than in rural areas (35.5%). Most of the dogs were male (65.9%), whereas only 34.1% were females. According to age classes, they were similarly distributed between young (51.3%) and adult (48.7%), ranging from two to 240 months.

### 3.2. Molecular Analysis

Out of 273 analyzed dog blood samples, 53.8% (147/273; 95%CI: 48–60%) showed positive results for *H. canis*, followed by 9.2% of the samples showing positive for *A. phagocytophilum*/*A. platys* (25/273; 95%CI: 6–13%), 3.3% for *B. canis rossi* (9/273; 95%CI: 2–6%), 2.6% for *E. canis* (7/273; 95%CI: 1–5%), 1.8% for *B. burgdorferi* s.l. (5/273; 95%CI: 1–4%) and 0.7% for *Rickettsia* sp. (2/273; 95%CI: 0–3%). As regards coinfections, out of 147 animals positive for *H. canis*, 17 were also infected with at least one bacterial pathogen, while three out of the seven dogs positive for *E. canis* were coinfected with another bacterial species (i.e., one each with *A. phagocytophilum*, *A. platys* and *B. burgdorferi* s.l.), and finally one animal was coinfected with *B. burgdorferi* s.l. and *A. platys*.

All samples positive for bacterial pathogens at PCR assays were purified, sequenced and analyzed with BLAST. Nineteen out of the 25 samples positive at the *A. phagocytophilum*/*A. platys* screening assay (Genbank: from ID OQ319089 to ID OQ319113) showed the highest identity percentage (99.81–100%) with analogous sequences of *A. phagocytophilum* strains, while six were more similar (99.66–100%) to *A. platys*. All nucleotide sequences obtained from the seven samples positive at PCR for *E. canis* DNA detection (Genbank: from ID OQ319077 to ID OQ319083) proved to be similar or identical (99.05–100%) to the *E. canis groEL* sequences published in the GenBank database. Three (Genbank: from ID OQ319086 to ID OQ319088) of the five samples positive for *B. burgdorferi* s.l. exhibited a nucleotide sequence with 99.63–100% similarity to those of *B. afzelii*, while the other two (Genbank: from ID OQ319084 to ID OQ319085) were identical to *B. burgdorferi* analogous sequences. Finally, out of the two samples positive for *Rickettsia* sp. during the first screening PCR, one (Genbank: from ID OQ319075) was identical to *gltA* sequences of *R. conorii* and the other (Genbank: from ID OQ319076) to *R. monacensis*.

Out of the 147 samples positive at real-time PCR for *Hepatozoon*, 88 were randomly selected and submitted to end-point PCR analysis, and all of them yielded positive results. Among them, 40 samples (Genbank: from ID OQ300438 to ID OQ300477) were Sanger sequenced and the BLAST analysis revealed 100% identity with sequences already present in Genbank^®^ and identified as *H. canis*. The nine samples positive for *B. canis* at real-time PCR were also submitted to end-point PCR, but only six samples resulted positive, while three gave a negative outcome, probably due to a limited amount of DNA present in the extracts. The sequences of the six samples (Genbank: from ID OQ300497 to ID OQ300502) showed 99.26% identity with a *B. canis rossi* isolate, reported in black-backed jackals in South Africa [29]. Considering that all results obtained at end-point PCR and sequencing were consistent with the identification of *H. canis* for the samples positive at real-time PCR with a melting temperature of 79/79.5 °C, the positive results obtained by the sole real-time PCR for the remaining samples (n = 59) were assumed as concordant to this identification.

### 3.3. Factors Influencing Pathogens’ Distribution

The differences in the proportion of positive animals to the different pathogens according to the considered risk factors are shown in Table 2. *Rickettsia* and *Borrelia* species were not displayed in this table, neither were their results analyzed through statistical analysis, in consideration of the very few positive samples.

The prevalence of the bacterial pathogens was similar in all subgroups for all factors investigated, apart from one factor for *A. phagocythophilum*, which was indeed more prevalent in the lowland agroecology (*p* = 0.032). On the contrary, protozoal pathogens (*H. canis* and *B. canis rossi*) showed significant differences for more than one factor. The prevalence of *H. canis* was higher in rural areas (*p* < 0.001) and in tick-infested dogs (*p* = 0.030). Concerning agroecology, dogs living in highland were significantly less infected by *H. canis* (*p* = 0.022), while the opposite was found for *B. canis rossi*, which showed the highest prevalence in the highland dog population (*p* = 0.001). Finally, younger dogs were more affected by *B. canis rossi* (*p* = 0.003).

The multivariable logistic regression model developed for *H. canis* (Table 3) confirmed the strong protective effect of living in an urban context and in highland.

The multivariable logistic regression model developed for *B. canis rossi* (Table 4) identified agroecology as the most influential factor on the presence of the parasite, with midland and highland at higher risk compared to lowland.

## 4. Discussion

In this study, 273 dogs from selected areas of Southern Ethiopia were studied using molecular methods with the aim of investigating the presence and distribution of bacterial and protozoal tick-borne pathogens with zoonotic potential and determining the associated risk factors.

A relevant finding of the study was the relatively high number of dogs (n = 19) that tested positive for *A. phagocytophilum*, a zoonotic emerging pathogen that causes Human granulocytic anaplasmosis (HGA). This finding was unexpected and at the same time alarming. An increase in human cases of this disease has been reported worldwide in recent years [30,31,32]. *A. phacocytophilum* is a multi-host pathogen characterized by a huge genetic variability influencing the host affinity of the bacterium. The isolates from dogs and humans are characterized by a strong similarity [33,34], and consequently dogs have been proposed as a sentinel for human infection risk [35]. Moreover, because of their proximity to humans, dogs may represent an amplifier and a source of infected ticks in a highly inhabited environment [35]. However, the enzootic cycle of this infectious agent appears complex and it is not completely clarified [36].

In Africa few studies have been conducted to detect the DNA of *A. phagocythophilum* in dog blood. The prevalence estimation (6.6%) of the present study was higher than what was recorded in previous surveys, conducted in many African countries including Algeria [37], Morocco [36], Ghana [38], Angola [39], South Africa [40] and other Sub-Saharan African countries [11], that reported positive findings ranging from 0% to 2.1%. However, the biomolecular methods and primers used differed among these studies. Other zoonotic strains of *Anaplasma*, genetically similar to *A. phagocytophilum,* were also detected in dogs in South Africa [41,42] and in Zambia [43]. Our observation is instead similar to those found in the dog populations of many European countries, such as Germany (5.7%), Romania (5.7% and 6.2%), Italy (6%) and Slovakia (8%). Similar values were reported in other countries worldwide, such as Turkey (6%), the United States (7.6%) and Brazil (7.1%). However, higher prevalence values were reported in both European and Asian countries: Hungary (11%), Poland (14%), China (13.2%, 10.9%, and 11.9%), Jordan (39.5%) and Iran (57.3%) [36]. In dogs, *A. phagocytophilum* is known as the causative agent for Canine granulocytic anaplasmosis (CGA) which can evolve asymptomatically or with few symptoms followed by a rapid positive evolution, or sometimes with an acute symptomatology characterized by fever, joint pain, gastrointestinal disorder and lethargy [44]. The dogs under study were asymptomatic and appeared clinically healthy at the moment of blood collection (even if mild or previous symptoms may have gone unnoticed), and therefore the detected prevalence can be considered remarkable.

Although *A. phaocytophilum* is primarily transmitted by ticks of the genus *Ixodes* in the USA (*I. scapulari, I. pacificus*), Europe (*I. ricinus*) and Asia (*I. persulacatus*), other tick species were also reported to play a role in its epidemiological cycle. In Africa some studies showed *A. phagocytophilum* in *Rhipicephalus sanguineus*, *Haemophysalis elliptica* and *Hyalomma* spp. [8,18,45,46]. In the dog population under this study, ticks belonging to the genus *Rhiphicephalus* and *Haemophysalis* were found, with the species *R. sanguineus* and *H. leachi* (unpublished data) highly represented. We can hypothesize the involvement of these tick species in transmission through their bite, but further studies should be performed to better understand the role of these ectoparasites.

The observation of 2.2% *A. platys* in dogs in the present study is also relevant from a public health perspective. This pathogen has occasionally been reported as a zoonotic agent [47,48,49]. Unlike with *A. phagocytophilum*, dogs are considered the natural host for *A. platys,* the causative agent for canine cyclic thrombocytopenia [50]. *Rhipicephalus sanguineus* is also implicated as the main competent vector of this bacterium, but DNA of *A. platys* has been detected in other tick species. *Anaplasma platys* is distributed worldwide and has been detected in several dog populations in African countries, such as Algeria [37], Morocco [36], Egypt [45,51], Green Cape [52], Ivory Coast [53], Kenya [53], Uganda [17], Angola [54], Zambia [43] and South Africa [42], with the percentage of positivity changing in relation to the geographical area and the type of animal population considered [11]. The results of the present study demonstrated the circulation of this pathogen in dogs in Ethiopia. It was found to have a higher prevalence in urban dogs than in rural animals, although not in a significant way. As suggested by Heylen et al. [11], this could be due to the behavior of *R. sanguineus* ticks, which prefer man-made constructions to hide in cracks and crevices to oviposit or molt and consequently infest dogs near these structures. The same consideration could be made for *Ehrlichia canis*, transmitted mainly by *R. sanguineus* and with the dog as the primary host. It causes canine monocytic ehrlichiosis [55] in dogs, which is characterized by clinical, subclinical and chronic phases. Since the sampled animals were asymptomatic, the positive dogs were probably in the chronic phase of infection. Although the prevalence was lower than those obtained in other African studies [11,40,42,45,56], the detection of this pathogen deserves attention not only because of the disease’s consequences for dogs’ health, but also because of the occasional reports of *E. canis* acting as a zoonotic agent [57,58].

Blood samples also tested positive for other important zoonotic bacteria belonging to *Borrelia burgdorferi* s.l. complex (of the Lyme group borreliae) and Spotted Fever Group (SFG) rickettsiae. *Borrelia burgdorferi* is the causative agent of Lyme disease, which is the most prevalent tick-borne zoonoses in the United States and Europe [59], but which is poorly studied in Africa although infection has been observed in humans [60,61]. Dogs are occasionally infected with this pathogen, which can cause symptoms including fever, general malaise, polyarthritis, lameness and lymph node enlargement [62]. The dog, sharing the same environment as humans, is considered a good indicator of the risk for people to become infected. The circulation of *B. burgdorferi* has been ascertained in dogs in Egypt both directly by molecular methods and indirectly by serological testing, and has been found in *R. sanguineus* collected from an infected dog [19,51]. On the other hand, this pathogen was not found in the ticks of dogs in South Africa [18]. The genospecies *B. afzelii* and *B. burdorferi* detected in this study, along with *B. garinii*, are the main causative agents of Lyme borreliosis in Europe and the US and are linked to different symptoms in humans [59]. To the best of the authors’ knowledge, this is the first report of *B. afzelii* circulation in Africa. Unlike *B. burgdorferi*, which causes primarily arthritis, this genospecies is associated mainly with Acrodermatitis Chronica Atrophicans (ACA) in humans. Our data thus provide interesting information on the circulation of this zoonotic pathogen. However, further studies should be conducted to obtain an idea of the real spread of borreliae in different African environments so as to determine the species of hosts and vectors that play a role in its epidemiological cycle.

Two dogs tested positive for *R. conorii* or *R. monacensis*, bacterial species belonging to the SFG. *Rickettsia conorii*, which is the causative agent of the human Mediterranean Spotted Fever (MSF), distributed mostly in the area surrounding the Mediterranean Sea and in Sub-Saharan Africa. MSF is the most important tick-borne disease occurring in North Africa [63]. This pathogen is transmitted primarily by *R. sanguineus*, which is considered the reservoir. Infected dogs are usually asymptomatic, but febrile illness has been observed [64]. DNA of *Rickettsia* spp. was reported in *H. elliptica* and *R. sanguineus* ticks from dogs in South Africa [8], in Ghana [65] and in Algeria [56], and in the blood of dogs in Nigeria [14] and in Angola [66]. *Rickettsia monacensis* causes MSF-like illness and is an emerging human pathogen. It was detected in ticks from North Africa (Morocco, Tunisia, Algeria), but it was rarely reported from dogs. It was detected by molecular methods in one animal in Green Cape [52] and in different species of ticks (i.e., *R. sanguineus*, *Ixodes boliviensis*, *I. ricinus*) collected from dogs living in different parts of the world, such as the United States [67], Costa Rica [68] and Romania [69]. Our data confirmed the circulation of SFG rickettsiae in Ethiopian dogs, and, since the dog is usually considered a sentinel of rickettsial infections in humans, other studies would be recommended to understand the real spread of these pathogens, their epidemiology and the risk for people to contract infection.

Piroplasms are probably widely present in African countries, although data are scant and information on their circulation is mostly limited to South Africa [70,71,72]. In the surveyed dog population, a large percentage of samples (53.8%) tested positive for *H. canis*, indicating that this pathogen is commonly found in the Ethiopian territory. The sequences obtained from positive samples of the study area were found to be very similar to specimens isolated from the black-backed jackal (*Canis mesomelas*) in South Africa [72], from the grey wolf (*Canis lupus*) in Germany [73] and from the dog in India [74]. The presence of this piroplasm in several African countries was already documented. For instance, in a study conducted in Sudan [20], 42.3% of dogs were positive for *H. canis* and in a subsequent study in Nigeria 20.3% of samples were positive for the same parasite [75]. These studies confirmed the tendency of this parasite to circulate in African dog populations with medium or high prevalence values, as in our study area. Moreover, areas at lower altitudes (i.e., lowlands and midlands) and located in a rural context showed a significantly higher prevalence of *H. canis* (71.8%). This could be explained by the fact that these areas constitute a more suitable environment for the survival and development of different stages of tick species acting as vectors of *H. canis*.

Concerning the *Babesia* spp. detected, in our study a small percentage (3.3%) of animals tested positive, and these samples were identified as *B. canis rossi*. This subspecies is particularly virulent for dogs and commonly reported in Sub-Saharan Africa [71], where the black-backed jackal was identified as a reservoir of *B. canis rossi* due to the high prevalence of the parasite in this host [70]. Indeed, nearly a third of jackals (29.7%) were infected by *B. canis rossi* in South Africa. Black-backed jackals are generally distributed in Northeast Africa, from Somalia and Eastern Ethiopia southward to Tanzania, and in Southwest Africa, from Southwestern Angola and Zimbabwe to the Western Cape Province in South Africa [70]. It is therefore reasonable to suspect that black-backed jackals are also responsible for maintaining *B. canis rossi* infection in Ethiopia, and specifically in the study area, since they are found in the nearby Nech Sar National Park [76]. This hypothesis is supported by its significantly higher prevalence in rural areas, where dogs share the same environment as wild canids and have a major risk of coming in contact with ticks and their pathogens. It is, however, more difficult to interpret why the prevalence of *B. canis rossi* was significantly higher with altitude, an opposite trend compared to the other pathogens. Finally, its higher prevalence in young dogs was probably related to the dog’s immunity, since acquired immunity to *B. rossi* increases with age, as has been observed under experimental conditions [77].

The use of real-time PCR for piroplasms was extremely useful in screening the canine blood samples and in identifying the positives through the melting curve analysis. This diagnostic approach represented an effective and cost-saving way to diagnose piroplasms using molecular technology. However, the use of the same primers for the detection of different protozoa belonging to the same group may represent a limitation of the study, since the species with higher parasitaemia may hide the presence of the other species, resulting in an underestimation of the prevalence of pathogens circulating at a lower level.

## 5. Conclusions

The more relevant finding of our study is the detection of different species of tick-borne pathogens with zoonotic potential (i.e., *A. phagocytophilum*, *B. afzelii*, *B. burgdorferi* ss, *R. conorii* and *R. monacensis*) and in particular the relatively high prevalence of *A. phagocytophilum*. These results demonstrate the circulation of the above-mentioned pathogens in the area, calling for a sensitization of the health authorities and medical personnel. The role of dogs in the epidemiological cycle of these pathogens needs further investigation. Furthermore, the tick species acting as vectors for *A. phagocytophilum* and *B. burgdorferi* s.l., pathogens usually vectored in temperate climates by *Ixodes* sp. (absent in this area), should be investigated.

The research enabled the determination of the occurrence of tick-borne pathogens relevant to dog health, revealing the wide diffusion of the relatively harmless *H. canis* and the sporadic presence of other more pathogenic species, such as *B. canis rossi, E. canis* and *A. platys*. Rural dogs seem at higher risk for most of the pathogens, probably due to their shared environment with other wild canids which can act as carriers of pathogens and spreaders of infected ticks (e.g., black-backed jackal for *Babesia*). However, the large majority of the investigated dogs have a lifestyle at risk, since most of the urban dogs are kept outdoors and left to free roaming during the day.

In conclusion, human and veterinary services should pay higher attention to these diseases, whose importance is underestimated in the area, probably due to the difficulties of performing an appropriate diagnosis in both humans and animals in such a context, which is characterized by poor diagnostic facilities.

## Figures and Tables

**Table 1 tropicalmed-08-00102-t001:** Primers used in the analyses performed for the different pathogens targeted by the study and expected amplicon length.

Pathogen	Primers	Expected Amplicon Length	Ref.
*Anaplasma phagocytophilum* *Anaplasma platys*	F (Ephpl-569F) ATGGTATGCAGTTTGATCGC R (Ephpl-1193R) TCTACTCTGTCTTTGCGTTC	624 bp	[23]
*Ehrlichia canis*	F (Ecan-163S) AAATGTAGTTGTAACGGGTGAACAG R (Ecan-573AS) AGATAATACCTCACGCTTCATAGACA	410 bp	[24]
*Borrelia burgdorferi* s.l.	F (GF) TACGATTTCTTATGTTGAGGG R (GR) CATTGCTTTTCGTCTATCACC	310 bp	[25]
*Rickettsia* spp.	F (Rp877p) GGGGGCCTGCTCACGGCGG R (Pp1258n) ATTGCAAAAAGTACAGTGAACA	381 bp	[26]
Piroplasms (real-time PCR and end-point PCR)	F 5′-CCAGCAGCCGCGGTAATTC-3′ R 5′-CTTTCGCAGTAGTTYGTCTTTAACAAATCT-3′	373 bp	[27]

**Table 2 tropicalmed-08-00102-t002:** Distribution of positivity in the investigated dog population according to different factors.

			*Hepatozoon* *canis*		*Anaplasma phagocythophilum*		*Babesia* *canis rossi*		*Ehrlichia* *canis*		*Anaplasma platys*	
Factor	Variable	Tested	Prev. *	*p*-Value	Prev. *	*p*-Value	Prev. *	*p*-Value	Prev. *	*p*-Value	Prev. *	*p*-Value
Sex	female	92	53.3%	>0.1	4.3%	>0.1	6.5%	=0.065	3.3%	>0.1	2.2%	>0.1
	male	181	54.1%		8.3%		1.7%		2.2%		2.2%	
Age class	young	140	51.4%	>0.1	5.7%	>0.1	**6.4%**	**=0.003**	2.9%	>0.1	2.1%	>0.1
	adult	133	56.4%		8.3%		**0.0%**		2.3%		2.3%	
Life style	mixed	14	50.0%	>0.1	7.1%	>0.1	0.0%	>0.1	7.1%	>0.1	0.0%	>0.1
	outdoor	259	54.1%		6.9%		3.5%		2.3%		2.3%	
Agroecology	lowland	186	**56.5%**	**=0.022**	**9.7%**	**=0.033**	**0.5%**	**=0.001**	3.8%	>0.1	2.7%	>0.1
	midland	40	**62.5%**		**0.0%**		**7.5%**		0.0%		0.0%	
	highland	47	**36.2%**		**2.1%**		**10.6%**		0.0%		2.1%	
Location	urban	176	**45.5%**	**<0.001**	8.5%	>0.1	1.7%	=0.072	4.0%	=0.053	3.4%	=0.066
	rural	97	**69.1%**		4.1%		6.2%		0.0%		0.0%	
Tick infestation	neg	172	**48.8%**	**=0.030**	7.0%	>0.1	3.5%	>0.1	2.9%	>0.1	2.3%	>0.1
	pos	101	**62.4%**		6.9%		3.0%		2.0%		2.0%	
Total		273	53.8%		7.0%		3.3%		2.6%		2.2%	

* prev. = prevalence. Significant differences (*p* < 0.05) at statistical analysis are highlighted by bold characters.

**Table 3 tropicalmed-08-00102-t003:** Results of the logistic regression model for *H. canis* (Hosmer–Lemeshow test: *p* = 0.880).

Factor	Variable	Odds Ratio	95% CI per Odds Ratio		*p*-Value
			Lower	Upper	
Location	rural	reference			
	urban	0.283	0.144	0.555	<0.001
Agroecology	lowland	reference			
	midland	0.478	0.195	1.171	>0.050
	highland	0.404	0.203	0.806	0.010

**Table 4 tropicalmed-08-00102-t004:** Results of the logistic regression model for *B. canis rossi* (Hosmer–Lemeshow test: *p* = 0.903).

Factor	Variable	Odds Ratio	95% CI per Odds Ratio		*p*-Value
			Lower	Upper	
Agroecology	lowland	reference			
	midland	19.207	1.801	204.900	0.014
	highland	17.935	1.961	164.063	0.011

## Data Availability

The files containing the data supporting our findings can be requested directly to the corresponding author.

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
