# Peer review of "Epidemiological Survey on Tick-Borne Pathogens with Zoonotic Potential in Dog Populations of Southern Ethiopia"

_tropicalmed, 2023, doi:10.3390/tropicalmed8020102_

Round 1
Reviewer 1 Report
Dear Sir
The article has been well prepared and nicely presented but need improvement in the following aspects
Abstract:
Line 28 to 30 "The present study contributes to increase the 28 knowledge on the epidemiology of tick-borne pathogens relevant for human and animal health in 29 an area poorly investigated so far" need revision
Introduction
Well prepared and the lines 61 and 62 "Like many African countries, the rate of urbanization in Ethiopia is increasing rapidly and it’s closely linked with human and dog populations" need modification
Materials and methods
The samples have been collected from the different places and a brief information of the general management practices may be given. Mostly dogs reared in closed environment have been considered, is there any street dogs in the studied locality, if so studying those will also helpful for detailed study and more incidence of the diseases.
Discussion
Neatly presented and connected to the objectives
Conclusion
It may be reduced and novel findings in one paragraph may be given
kind regards
Author Response
The article has been well prepared and nicely presented but need improvement in the following aspects
Abstract:
Line 28 to 30 "The present study contributes to increase the 28 knowledge on the epidemiology of tick-borne pathogens relevant for human and animal health in 29 an area poorly investigated so far" need revision
REPLY: We modify the sentence to improve its clarity.
Introduction
Well prepared and the lines 61 and 62 "Like many African countries, the rate of urbanization in Ethiopia is increasing rapidly and it’s closely linked with human and dog populations" need modification
REPLY: We modify the sentence to improve its clarity.
Materials and methods
The samples have been collected from the different places and a brief information of the general management practices may be given. Mostly dogs reared in closed environment have been considered, is there any street dogs in the studied locality, if so studying those will also helpful for detailed study and more incidence of the diseases.
REPLY: In the study area, owned dogs are mostly kept always outdoor (see table 3) and left to free-roaming during the day, resulting in a lifestyle comparable to that of the stray dogs and at high risk for TBD. Stray (or street) dogs are non-owned dogs free-roaming the whole day, also during the night. In the study area, such as in the whole country, the safe containment and management of stray dogs is extremely difficult and expensive (it usually needs tele-anesthesia), due to the risk of rabies and dog bite. This is now specified in the revised manuscript, along with a brief description of the general management practices, mainly to clarify the similarity in lifestyle between owned and stray dogs.
Discussion
Neatly presented and connected to the objectives
Conclusion
It may be reduced and novel findings in one paragraph may be given
REPLY: We tried to delete unnecessary parts, but it was already focused on the main novel findings. The separation in two paragraphs is also important, in order to keep separated the findings related to public health from the ones related to dogs’ health.
Reviewer 2 Report
See the attached file

Author Response
Title
- S of southern with capital letter.
REPLY: corrected
Abstract
- Write significant values of the given results. Important results are missing in the abstract.
REPLY: A sentence on the more significant results of the multivariable analysis has now been added in the abstract, although the limit in number of words do not allow us a more detailed description here.
- Should be Hepatozoon (H.) canis, Anaplasma (A.) phagocythophilum, Babesia (B.) canis
REPLY: According to our experience the proposed formal way of writing the names is not common for parasites, while in the case of bacteria the abbreviated name is usually written into brackets the first time they are cited. We modified accordingly in the main text, while in the abstract we preferred to keep always the extended names.
Introduction
- Begin your Introduction by clearly identifying the subject area of interest. Establish the context by providing a brief and balanced review of the relevant published literature that is available on the subject
REPLY: A new sentence has been added at the beginning of the introduction to immediately identify the topic under study, and unnecessary parts of the first paragraph deleted to shorten the more general introductive part. References has been revised, also according to suggestions of Reviewer 3.
- Line 41: Avoid to start sentence with abbreviation
REPLY: corrected
Materials and methods
- What was the sampling method? How you calculate the sample size?
REPLY: The sampling method was through blood samples collection. Then, blood samples were applied to FTA card and conserved until analysis. We rearranged the whole paragraph on sampling method to improve its clarity and including two sentences on inclusion/exclusion criteria, to answer previous comments. Besides, the definition of the minimum expected sample size has been added in the revised manuscript.
- Line 85: Should be µL
REPLY: Corrected
- Line 97: Add protocol briefly (DNA isolation)
REPLY: This is a very standardized procedure common for many commercial kits, and we kept the sentence “according to the manufacturer’s instructions” since details are freely available at the production company website.
- Line 103: is it 0.3?
REPLY: Corrected
- Line 104: Should be Anaplasma (A.) phagocythophilum, etc
REPLY: See previous comment.
- Line 130: add protocol briefly (PCR protocol)
REPLY: a brief description of the protocol was added in the revised manuscript
Results
- Lines 157-160: add statistical values
REPLY: The reported lines (157-160) simply describe the dog population under investigation. Risk factors and related statistical values are shown in the paragraph "Factors influencing pathogens' distribution" below
- Table 2: data should be mentioned according statistical analyses
REPLY: The presented table 2 gives an overall view of data. Data according to statistical values are shown in Table 3, in which is also possible to see the statistically significant differences.
- Table 3: What about Chi-square values?
REPLY: We added in the footnote (Table 3) the statistical method we used to calculate differences (Pearson Chi-Square test or the Fisher exact when necessary)
- Table 3: comma (,) should be replaced with dot (.) correct it throughout the text
REPLY: Corrected
- Table 5: why another factor location is missing in this table?
REPLY: Tables 4 and 5 show the results of the logistic regression models for H. canis and B. canis rossi based on a backward steps approach, and therefore all variables were included at the first step, but only the ones included in the best model are displayed in the table. Concerning B. canis rossi, the model with only the agroecology variable resulted the best one, therefore location variable is missing (such as the other variables).
Discussion
- Line 257: stand for? at least once write it complete
REPLY: Rhipicephalus. Corrected
Reviewer 3 Report
The present study investigates the occurrence and the prevalence of bacterial and protozoal tick-borne pathogens using molecular methods in owned dogs in Southern Ethiopia, a place with few studies available on this topic. The more relevant finding of the study is the detection of different species of tick-borne pathogens with zoonotic potential, which is important from a public health perspective. This makes it highly relevant and worthy to publish.
Comments
-Sequences must be available so that others can replicate the analysis or used them in other studies, please submit the sequences to GenBank and provide accession numbers.
-Please verify or find more appropriate references, specially for the introduction section. Non appropriate references include ref 11, 12, 18, 19, 21.
-No coinfection was found? it could be discussed.
-Line 48. Include the whole name – Vector Borne Diseases (VBDs)
-Line 50. Not sure what conductive climate means…
-Line 105 and throughout the text. Replace B. burgorderfi sl with B. burgorderfi s.l.
-Line 120. Replace usingan for using an
-Table 1. Merge the two last rows.
-Line 146. Please specify what are the differences between agroecology (i.e., highland, midland, lowland), altitude? What altitude exactly? Is the vegetation type different? How?
-Line 183. Separate “sequencedand”
-Line 183-184. 96.7-100% identity is a big range, 96% is not high, why are you getting these results? Please explain.
Author Response
The present study investigates the occurrence and the prevalence of bacterial and protozoal tick-borne pathogens using molecular methods in owned dogs in Southern Ethiopia, a place with few studies available on this topic. The more relevant finding of the study is the detection of different species of tick-borne pathogens with zoonotic potential, which is important from a public health perspective. This makes it highly relevant and worthy to publish.
Comments
-Sequences must be available so that others can replicate the analysis or used them in other studies, please submit the sequences to GenBank and provide accession numbers.
REPLY: Sequences of the positive samples were deposited in GenBank and their numbers are provided in the revised manuscript in the paragraph “3.2. Molecular analysis”.
-Please verify or find more appropriate references, specially for the introduction section. Non appropriate references include ref 11, 12, 18, 19, 21.
REPLY: We modified ref 11, 12 and 21, as per reviewer suggestion. Concerning ref 18 and 19, these are actually studies reporting the occurrence of “bacterial pathogens” (other than Rickettsia, Anaplasma and Ehrlichia) in African countries (i.e., Coxiella and Borrelia) and therefore we consider these references appropriate in this part of the manuscript.
-No coinfection was found? it could be discussed.
REPLY: Considering the high number of pathogens investigated, it was difficult to interpret the occurrence of coinfections, that were anyhow not so frequent. A sentence was added in the results section to report the coinfections detected.
-Line 48. Include the whole name – Vector Borne Diseases (VBDs)
REPLY: The whole name was added
-Line 50. Not sure what conductive climate means…
REPLY: “Conducive” (not conductive) means favorable. In this case a favorable/conducive/beneficial climate for arthropods diffusion.
-Line 105 and throughout the text. Replace B. burgorderfi sl with B. burgorderfi s.l.
REPLY: It was corrected throughout the text.
-Line 120. Replace usingan for using an
REPLY: corrected.
-Table 1. Merge the two last rows.
REPLY: done
-Line 146. Please specify what are the differences between agroecology (i.e., highland, midland, lowland), altitude? What altitude exactly? Is the vegetation type different? How?
REPLY: We are grateful to reviewer for highlighting this point. We introduced a brief description of the geo-climatic characteristics of the whole study area and of the three agroecologies, at the beginning of the paragraph 2.1 sampling.
-Line 183. Separate “sequencedand”
REPLY: corrected
-Line 183-184. 96.7-100% identity is a big range, 96% is not high, why are you getting these results? Please explain.
REPLY: we agree with Reviewer’s comment and we checked all our sequences, also because of the concomitant request for submission in Genbank. We realised that chromatograms were not thoroughly corrected in the previous version, due to the short time available before the deadline for manuscript submission that pushed us to work a bit too much in a rush. We are sorry for this. Now, we revised accurately all sequences correcting them, and the percentage of identity improved for all isolates reaching 100% for H. canis and 99.26% for B. canis rossi (see deposited sequences for details). We revised and corrected also sequences for bacterial pathogens and the percentage of identity improved a bit also for some of them.
Round 2
Reviewer 2 Report
Line 102: On what basis were authors decided to examine only 276 animals?
What was the sampling method or type? Simple random, convenient, etc.?
Line 198: what is the correct sample size? 276 or 273.
Line 105: Authors declared districts of province as a variable but did not describe their association pathogens statistically.
Table 3 only describes the location (rural, urban) and acarology (highland, midland, lowland), but not districts.
So Table 2 data should be presented with statistical analyses. Or remove this variable and Table 2 from the manuscript.
Line 189: Did you use Chi-square and Fisher’s exact tests on all variables? If yes, justify it and if not, mention clearly, which test applied to which variable.
In Table 3, instead of N pos. values, add Chi-square value
Author Response
Line 102: On what basis were authors decided to examine only 276 animals?
REPLY: The parameters set to define the expected sample size were introduced in the last revision (lines 93-96 in the R2 manuscript, where we changed ‘minimum’ with ‘expected’). The most commonly used sample size in cross-sectional study is 385, that is based on parameters similar to ours, but with a 5% of accepted error. Being aware of the logistic and practical difficulties in collecting blood samples from dogs in the study area (blood sampling in dogs is very uncommon for both pet owners and veterinary staff), we decided to set a lower sample size, keeping anyhow a good precision (accepted error was increased only to 6%). During the sampling activity, 297 dogs were visited but for 21 of them it was not possible to collect blood, due to the aggressive behavior of the dogs and consequent risk for the operator. Blood samples were collected from 276 dogs, but 3 samples were not exploitable, because FTA cards were ruined during the transport to Italy. This difference in expected and real sample size is now explained at the beginning of the Results section (lines 185-187).
What was the sampling method or type? Simple random, convenient, etc.?
REPLY: We probably misunderstand the comment from the first revision. First, the areas (districts) to be investigated were selected by the Gamo Zone administration, choosing the ones with higher dog population, based on their experience with rabies vaccination. However, the households owning a dog and the number of dogs present in each area were not exactly known. Therefore, we went to each kebele/village of the four districts with a local vet assistant which was known by the villagers (to create trust) and the households for sample collection were opportunistically selected by asking them if they had dogs. These aspects are now better specified in the revised manuscript.
Line 198: what is the correct sample size? 276 or 273.
REPLY: See answer to previous comment.
Line 105: Authors declared districts of province as a variable but did not describe their association pathogens statistically.
REPLY: Actually, all information collected during sampling activity are reported at line 105 (now line 97), while the variables included in the statistical analysis are reported in the ‘Data analysis’ paragraph at lines 172-174, and correspond to the ones in table 3. The district is based on administrative boundaries and therefore we don’t consider it as a factor that can influence the occurrence of pathogen. For this reason, we used location and agroecology only in risk factors analysis, considering that these aspects are potentially influencing the presence of ticks and transmitted pathogens.
Table 3 only describes the location (rural, urban) and acarology (highland, midland, lowland), but not districts.
REPLY: See answer to previous comment.
So Table 2 data should be presented with statistical analyses. Or remove this variable and Table 2 from the manuscript.
REPLY: The aim of table 2 was to present in details and with transparency the provenance and individual characteristics of sampled animals, but it didn’t show any results of the molecular analysis. However, based on reviewer comment, we understand that the content of the table is unclear and it may be misleading. Therefore, we accept the suggestion to delete it, moving the more relevant information in the main text (lines 187-195 in the R2 manuscript). Obviously, the number of the following tables has been rearranged accordingly.
Line 189: Did you use Chi-square and Fisher’s exact tests on all variables? If yes, justify it and if not, mention clearly, which test applied to which variable.
REPLY: Both tests have been used for all variables, when possible (Fisher test was used only for 2x2 table – factors with only two variables). The results are usually similar, but Fisher's exact test is more accurate than the chi-square test when the expected numbers are small (usually under 5 in one of the cells), and therefore we used the p-value obtained by the former in these cases. We slightly modified the sentence in the M&M section, and we inserted the p-value of the more appropriate test in the Table 3 (now Table 2), as requested in the following point.
In Table 3, instead of N pos. values, add Chi-square value
REPLY: we agree with the reviewer suggestion that the p-value of the statistical test may be more informative and therefore we deleted the number of positive (it can be deduced by the prevalence value) and inserted the p-value, keeping the bold character to facilitate an immediate perception of which differences were significant. Be aware that track change option was not maintained during this modification to avoid an excessive number of highlighted revisions.
Round 3
Reviewer 2 Report
Accept in present form